# Effect of Temperature on the Electrical and Thermal Behaviour of a Photovoltaic/Thermal System Cooled Using SiC Nanofluid: An Experimental and Comparison Study

Hussein A. Kazem [1,2,*], Miqdam T. Chaichan [3], Ali H. A. Al-Waeli [4], Hasila Jarimi [2], Adnan Ibrahim [2] and K. Sopian [2]

1   Faculty of Engineering, Sohar University, P.O. Box 44, Sohar 311, Oman
2   Solar Energy Research Institute, Universiti Kebangsaan Malaysia, 43600 Bangi, Selangor, Malaysia
3   Energy and Renewable Energies Technology Research Center, University of Technology, Baghdad 19006, Iraq
4   Engineering Department, American University of Iraq, Sulaimani, Kurdistan Region, Sulaimani 46001, Iraq
*   Correspondence: h.kazem@su.edu.om; Tel.: +968-2672-0101

**Abstract:** A photovoltaic/thermal system (PV/T) was investigated experimentally using silicon carbide nanofluid as a cooling fluid. A PV/T system was tested in Oman with 0.5 wt.% of nanoparticles in terms of thermophysical properties, performance parameters, and efficiencies. At 25 °C, it was found that there is an increase in the fluid's thermal conductivity, density, and viscosity up to 6.64%, 13%, and 12%, respectively. When examining the effect of increasing the density and viscosity (by adding nanoparticles to the base fluid) on the pumping power, it was found that using turbulent flow reduces the required pumping force and vice versa for the case of laminar flow. The electrical efficiency was enhanced by up to 25.3% compared with the conventional PV module and the thermal efficiency by up to 98.6% compared with the water-cooling PV/T system. The results were compared with the literature in terms of cooling methods, nanoparticles, and similar studies that used SiC nanofluid. The results and comparison of this study are useful for engineers and researchers interested in nanofluid cooling of PV/T systems. The study aims to facilitate the task of engineers and designers of photovoltaic plants in Oman to obtain the best means to overcome the effects of high solar radiation intensity and high ambient temperatures and the best PV/T systems for this purpose.

**Keywords:** PV/T system; nano-SiC; stability; pumping power; thermophysical properties

## 1. Introduction

There are many sources of renewable energy, but solar energy has become more attractive in the last few decades. The sunshine daily sends light and heat to the Earth. Solar light is utilized by solar photovoltaics (PV) to generate electricity [1], while thermal collectors utilize solar thermal energy [2]. Solar thermal energy is collected using solar collectors for many applications such as solar water heaters, air heating, and concentrating systems to generate electricity, etc. [3]. Combining PV and thermal collectors will produce a photovoltaic/thermal (PV/T) system. The PV/T system will reduce the located area and improve the system efficiency [4,5]. Many research studies have been conducted to investigate PV/T systems in terms of technical, economic, environmental, and social aspects. PV/T systems could be classified based on the type of cooling fluid, collector configuration, etc. [6]. Furthermore, the PV/T fluid will reduce the PV temperature and improve its electrical efficiency. Many studies published in the literature have investigated the PV/T performance theoretically, experimentally, numerically, or a mix of approaches [7,8].

Many researchers and specialists have proposed and created new and different methodologies and strategies to evaluate the performance of PV/T systems. A concise survey of these strategies is exhibited here. Sivamurugan et al. [9] designed and developed a hybrid collector in which a PV panel is cooled with a nanofluid formed from manganese

oxide–water. The electrical and thermal efficiencies of the system were examined when the nanofluid was circulated at the flow rates of 0.5 and 1.0 LPM. The maximum thermal efficiencies reached 48.1% and 53.8% when the nanofluid was circulated at volume flow rates of 0.5 and 1.0 LPM, respectively. The highest electrical efficiencies reached were 18.32% and 19.35%, respectively, for the same two flow rates. Kim et al. [10] added silicon carbide/indium tin oxide (SiC/ITO) to water to be used as a working coolant in a PV/T system. The researchers used the optical transmittance at different mixing ratios to determine the optimal mixing ratio of the SiC and ITO nanofluid. The hybrid liquid nanofluid showed a maximum photothermal efficiency of 34.1%, which is 38.7% higher than that resulted from using nano-SiC–water nanoparticles. Elangovan et al. [11] studied the effect of adding nano-TiO2 to water on the temperature of the PV module and compared it to cooling the module with water. The thermal efficiency of the nanofluid-cooled system reached 48.38% and 54.03% at flow rates of 0.5 l/m and 1.0 l/m, respectively. The highest electrical efficiencies of the PV/T system were 18.32% and 19.35% for volume flow rates of 0.5 l/m and 1.0 l/m, respectively. In addition, Menon et al. [12] used water/nano-CuO nanofluid to cool an unglazed PVT system using a thermal collector made of serpentine sheets and tubes. While the temperature of the PV panel reached 68.4 °C (at noon), the temperatures of the PV panels in the water and nanofluid-cooled PV/T systems were reduced by 15 °C and 23.7 °C, respectively. The electrical efficiencies of the water and nanofluid-cooled PV/T systems were also increased by 12.32% and 35.67% compared to conventional PV modules, respectively. The overall efficiency of the nanofluid-cooled PV/T system was above that of the water-cooled PV/T system by 21%.

The execution of a PV/T system that used nanofluid as heat transducer media has been explored by Matin Ghadidri et al. [13]. It was announced that the overall system productivity is approximately 76% using the nanofluid for cooling with 3 wt.%. Michael et al. [14] have considered the use of 0.05 wt.% copper oxide–water nanofluid flow in copper pipes of the PV/T system. The authors claimed that the thermal productivity was 45.76%. Adun et al. [15] used a new type of liquid (trinary nanofluids), which is a composite of three types of nanoparticles added together in the base liquid. The researchers used nano-Al2O3-ZnO-Fe3O4 added to water. The results showed that the concentration of trinary nanoparticles best suited for both laminar and turbulent flows was 0.5% by volume. The highest electrical and thermal efficiencies achieved were 13.75% and 59.38%, respectively. The PV panel temperature of the trinary PV/T system was decreased by 8.81 °C. The performance of the hybrid PV/T system was investigated by Ying-Ying Wu et al. [16]. The results show that the unglazed PV/T system delivers higher performance using nanofluid. To maximize the productivity of the flat plate hybrid collector, Srimanickam [17] used two types of coolants (water and nanofluids). The nanofluid was prepared from nano-Al2O3 (0.1% vol.) added to water. Four volume flow rates (0.5, 1.0, 1.5, and 2.0 LPM) were tested. The best cooling effect of the PV/T system was achieved when using a flow rate of 2.0 LPM. At the time, the conventional PV temperature reached its maximum value of 68.2 °C, and the temperature of the nanofluid-cooled PV/T system was 63.8 °C. The flat platform PV/T system types are presented by Fudholi et al. [18], in which collectors and their performance are evaluated with water, air, and both. Mortada and Hussein [19] experimentally studied the effect of using coolants such as water or aqueous nano-Al2O3 in a PV/T system. Various concentrations of nanoparticles and flow rates were tested. The nanofluid with a concentration of 3% of nano-Al2O3 achieved the highest reduction in the surface temperature of the PV module and was about 23.14% compared to the conventional PV one. At turbulent flow, the electrical efficiency increased by 20.2% compared to laminar flow (15%).

Sopian et al. [20] developed a PV/T design that significantly enhanced the thermal efficiency of the PV/T system after comparing the traditional PV/T system with the building integrated PV/T (BIPV/T) system. BIPV/T systems were found to produce better efficiencies than traditional PV/T systems. The system produced higher thermal energy and was largely affected by the cost, leakage, and light weight of the system.

Thakare et al. [21] developed a system for evaluating the PV/T system in stagnant water retention. In the case of active water flow, the best mass flow rate found was 235 ml/h. The thermal energy efficiency found was 49.68%. On the other hand, the water's low density of the mass flow at 173 ml/h was able to generate more efficient results. The performance of thermal energies was measured at 51.27%, and the electrical performance was 12.54%. The authors concluded that active water flow would increase thermal efficiency compared to conventional methods.

Kazem et al. [6] conducted a comparison study for three different PV/T configurations (direct, web, and spiral flow) in Sohar, Oman. The study compared the experimental and simulation (using COMSOL) results. Moreover, thermal and electrical performance were compared, evaluated, and discussed. The authors claimed that the spiral flow configuration gave the best electrical and thermal performances compared to the other two configurations. It was worth mentioning that water is used as a cooling fluid. Table 1 illustrates a comparison of the PV/T system using a different type of nanofluid (nanoparticles–water).

**Table 1.** The critical findings of using nanofluid in various PV/T applications.

| Nanoparticles | Critical Findings | Reference |
|---|---|---|
| MWCNTs | The increase in the volumetric concentration of MWCNT in the base fluid (SG/H2O) contributed to the improvement of the thermal and electrical conductivities of the used PV/T system. The researchers determined the optimal conditions to reach the best electrical and thermal efficiencies. These conditions are an operating temperature of 66.2 °C and a concentration of MWCNT of 0.125% by volume in the nanofluid. | [22] |
| $SiO_2$ | Nano-$SiO_2$ was added to water in variable volume ratios (0.1–0.2–0.3%). The researchers concluded that radiation, flow rate, and nano-$SiO_2$ volumetric concentration are the most influential factors in electrical and thermal efficiency. When comparing the experimental results with the results of the developed mathematical models for both electrical and thermal efficiency of the PV/T system, a good convergence was found between them. | [23] |
| $Fe_3O_4$ | Nano-$Fe_3O_4$ was added to the base fluid (water), and a vertical magnetic field was imposed on the flowing fluid to increase the cooling rate of the bimetallic system, thus improving its electrical and thermal efficiency. The addition of Nano-$Fe_3O_4$ lowered the PF unit temperature, which increased the electrical and thermal efficiencies by 0.05% and 0.39%, respectively. | [24] |
| $SiO_2$, $Al_2O_3$, ZnO, and CuO | Four types of nanoparticles ($SiO_2$, $Al_2O_3$, ZnO, and CuO) were added to water, and the produced nanofluids were used to cool the CPVT collector. The maximum increase in the heat transfer rate was when using water–nano-$SiO_2$ then water–nano-$Al_2O_3$, water–nano-ZnO, water–nano-CuO, and finally water. The researchers found that water–nano-$SiO_2$ transfers and dissipates heat better than other nanofluids due to its high thermal conductivity. | [25] |
| Graphene nanoplatelets (GNPs) | The study results show that the highest nanofluid stability was achieved at (1-1) SDBS-GNPs sample with 60 min of ultrasonication mixing. The thermal conductivity of nanofluid enhanced to 8.36% was compared with pure water, which has the lowest viscosity of 7.4%. | [26] |
| Mg-Ag | It was found that the hybrid nanofluid enhances the heat transfer. The study found that the prediction model shows high accuracy compared with the experimental results. | [27] |
| $Fe_3O_4$ | The authors investigated and compared a conventional PV module with a PV/T system in terms of efficiency improvement. It is found that PV/T efficiency is 52% compared with 4.4% for conventional PV. The PV/T efficiency increased up to 76% when $Fe_3O_4$ nanofluid was used with a 3% concentration. More efficiency enhancement was produced when a magnetic field was used with the nanofluid, where the efficiency improved to 79%. | [13] |
| $Al_2O_3$ | The nanofluid viscosity increased with the increase of the nanoparticles' volume fraction. The stability of the nanofluid that contained oleic acid was less impacted by the viscosity increment. | [28] |
| $Al_2O_3$ and CuO | The authors claimed that the stability of nanofluid is highly affected by the thermal conductivity and viscosity of the nanofluid. However, the stability increased with the decrease of viscosity, and the increase of nanoparticle mass fraction will decrease the stability. | [29] |



**Table 1.** *Cont.*

| Nanoparticles | Critical Findings | Reference |
| --- | --- | --- |
| Ag, SiO$_2$, and CNT | Multi-walled carbon nanotubes with SiO$_2$ and Ag were used in the investigation. The new mixture reduced the solar cell temperature and enhanced power production. The CNT and silica added to the basic silver nano-disc enhanced the absorption of the ultraviolet rays. | [30] |
| CuO and Al$_2$O$_3$ | The study reveals that cooling is enhanced when distilled water is used compared with ethylene glycol-base fluid. Moreover, the Cu nanofluid shows the highest electrical, thermal, and overall efficiencies compared with Al2O3–water and Al$_2$O$_3$–Ethylene glycol. | [31] |
| Al$_2$O$_3$ | The authors found that the increase in solar irradiance will increase the PV productivity but at the same time increase temperature, which reduces the productivity. The use of Al$_2$O$_3$ nanofluid cooled the PV and enhanced the productivity compared with pure water cooling. | [32] |
| SiC | The study investigated SiC nanofluid and pure water for PV/T cooling. It was found that at 3% weight of SiC, the viscosity, density, and thermal conductivity increased by 5.18%, 8.2%, and 4.3%, respectively. Moreover, it was found that there is an efficiency increase of 24% compared with conventional PV modules. Finally, it was found that the heat transfer efficiency increased to 100.19% in comparison to water cooling only. | [33] |
| CuO and Al$_2$O$_3$ | The economic effects of using CuO and Al$_2$O$_3$ nanofluids to cool PV/T systems were studied. When comparing the performance of the systems with those operating with conventional liquids, the results showed that the nanofluid-cooled PV/T systems have a lower recovery period, which results in better economic savings compared to the water-cooled systems. | [34] |
| SiC | In this study, an economic evaluation of a grid-connected PV/T system cooled with a water + nano-SiC nanofluid is evaluated. The annual production factor of the studied GCPVT system ranges from 128.34 to 183.75 kWh/kW. The energy cost of this system was 0.196 USD/kWh, the payback period was 7–8 years, and the efficiency was 14.25%. | [35] |
| MWCNT | In this study, the researchers used MWCNT water added as the cooling fluid in the PVT system. The addition of MWCNT caused an increase in the cooling of the PV/T system and a decrease in the temperature of the solar panel by about 12° C and also caused a clear increase in the thermal and electrical efficiencies, bringing the total efficiency of the system to about 83.26%. | [36] |
| SiC | In this study, three types of base liquids were prepared, which were water, water + 35% ethylene glycol, and water + 35% propylene glycol, to which nano-SiC and cetyltrichromyl ammonium bromide were added. The thermal conductivity of the prepared fluids was close, while the density and viscosity of glycol fluids were higher than water. The stability of nanoparticles in glycol suspensions was more than that of water when mixed with ultrasonic vibration for a period of 4 to 6 h. | [37] |
| MWCNT, Al$_2$O$_3$, and CuO | The use of nanofluids in the PV/T system caused a higher electrical and thermal output compared to cooling with water. The use of a nanofluid containing MWCNT and CuO caused a temperature reduction of about 19% in PV modules' temperatures. The electrical efficiency of PV/T systems operating with MWCNT, Al$_2$O$_3$, and CuO nanofluids increased by 60%, 55%, and 52% compared to conventional PV. | [8] |
| Nano-MXene (Ti$_3$C$_2$) | Nano-MXene (Ti$_3$C$_2$) with three mass fractions 0.05, 0.10, and 0.20% was added to water, and the produced nanofluid was used to cool the PV/T system. The thermal conductivity of the nanofluid when adding 0.20% mass fraction increased by 47% compared to water. The maximum electrical efficiency was 13.95%, and the maximum thermal efficiency was 81.15% when cooling with water/MXene nanofluid. | [38] |
| GNT, TiO$_2$, and SiO$_2$ | The PV/T system was cooled using nanofluid composed of graphene tubes, TiO$_2$, and SiO$_2$ nanoparticles. The maximum thermal efficiency obtained using graphene and water suspension was 89.11%. The highest increase in electrical efficiency was 24.15% compared to water cooling. | [39] |
| Fe$_2$O$_3$ | Nano-Fe$_2$O$_3$ was added to water and ethylene glycol to form a nanofluid that cooled the PEFT system. The addition of 2% of nanoparticles caused an increase in the thermal conductivity of the nanofluid by 140%, and this fluid has good stability. The highest overall system efficiency obtained was 72% higher than that of an independent single crystal PV system. When using polycrystalline PV, the highest overall efficiency increase was 77.65% compared to the standalone PV system. | [40] |
| SiC | Nano-SiC was added to water to form a nanofluid that cooled the nano-paraffin in a PV/T collector tank. The electrical efficiency of the system was greatly improved, and the electrical power output increased. The maximum electrical efficiency obtained was 13.7% compared to 7.11% for a conventional PV system. The temperature of the PV/T system panel was reduced to 39.52 °C compared to 68.3 °C for a conventional PV panel. | [41] |

From Table 1, it is found that the effect of the trend of using different nanoparticles is an increase in heat transfer and an improved cooling process, which lead to improved efficiency. Moreover, some nanoparticles used in cooling nanofluid show superior performance compared to others, such as single-wall carbon nanotubes, SiC, Cu, and $Al_2O_3$. Furthermore, the increase in nanoparticle mass fraction will affect the nanofluid physical properties and increase the cost and pump power consumption.

The current study aims to evaluate the PV/T system cooled using SiC nanofluid in Oman based on experimental data sets. A PV/T and conventional PV module were installed and tested in Sohar, Oman. The novelty of this study is to evaluate the electrical and thermal behavior of nanofluid-based PV/T systems in harsh weather conditions. Moreover, a different comparison of PV/T results has been conducted and presented concerning similar studies in the literature in terms of cooling methods by other nanofluids and SiC nanofluid. The results of the study will give a roadmap for engineers and designers to establish PV fields in Oman on the best method to overcome the effects of high solar radiation intensity and high temperatures and the best PV/T systems for this purpose.

This paper contains four sections as follows: (Section 1) Introduction, (Section 2) Experimental Setup, (Section 3) Results and Discussion, and (Section 4) Conclusions. The results section contains (Section 3.1) Thermophysical Properties, (Section 3.2) Experimental Results, and (Section 3.3) Comparison with Literature.

## 2. Experimental Setup

### 2.1. Sohar Metrological Data

The investigated systems were installed north of Oman near the coast in Sohar city at 24.3461 N latitude and 56.7075 E longitude, respectively. Sohar have a desertic weather and is hot in summer. Figure 1a–d show a twelve-months record of ambient temperature, precipitation, sunshine hours, and solar radiation, respectively [42,43]. The highest temperature reached 50 °C in summer. However, rainfall was found to be high in winter. In general, sunshine hours are relatively high in Oman, ranging between 8 and 12 h. Dust is suspended clearly in the air of Sohar, and its accumulation causes a decrease in the productivity of photovoltaic panels [42,43]. This factor is considered one of the most important difficulties that interfere with the results of the tests. To completely neutralize its effect on the photovoltaic panels, the photovoltaic panels were cleaned before sunrise in the morning for all days of the tests.

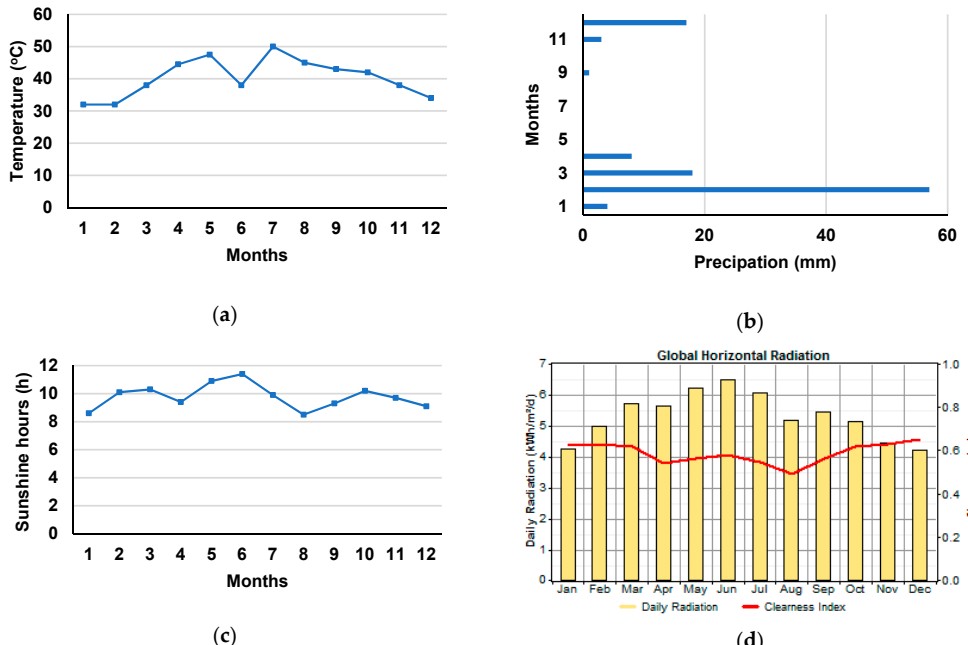

**Figure 1.** Monthly average of (**a**) ambient temperature, (**b**) rainfall, (**c**) sunshine hours, and (**d**) radiation.

## 2.2. Materials

In choosing the materials used in the experiments, the research team relied on several practical studies that have been conducted since 2016 until today in the study area. In order not to repeat the experiments, nano-SiC was chosen as the nanoparticles additive based on the results of [33,44]. The base fluid was chosen as the mixture of water + ethanol glycol by volume fractions of 75% water to 25% ethylene glycol based on the results of [45]. Cetyltrichromyl ammonium bromide (CTAB) was adopted as a surfactant added by 0.51 ml/l depending on the results of [37]. As for the added mass fraction of nano-SiC to the base fluid, it was chosen as 0.5% depending on the results of [46]. The used nano-SiC has high thermal conductivity of about 150 W/m K compared to the particle used in the previous reference, which was 40 W/m K). Nano-SiC particles were added to the base fluid and mixed for three and a half hours depending on [47,48]. After the mixing was completed, samples were taken from the prepared suspension, and thermophysical tests were performed on them. Table 2 lists the nano-SiC specifications used.

**Table 2.** SiC nanoparticles' specifications.

| Feature | Specification |
| --- | --- |
| Supplier | Zhenxin Ferroalloy Supplier (China) |
| External shape | Black powder |
| Purity | 98.8% |
| pH | 3.5–7.5 |
| Crystal form | Cubic |
| Particles size (nm) | 20–35 |
| Density (g/cm$^3$) | 3180 |
| Melting point (°C) | 2740 |
| Microhardness (kg/cm$^3$) | 3280 |
| Thermal conductivity (W/m K) | 125–167 |

After mixing the nanoparticles with the base fluid and surfactant, samples were taken from it to examine the thermophysical properties before starting the experiments. During the experiments and after operating for a whole day, a sample of the nanofluid was taken, and its thermal conductivity and stability were checked to ensure that this fluid did not lose the required basic properties. Despite the difficulty of this matter, such an unprecedented procedure in previous research is very important to practically ensure the quality of the results.

## 2.3. The PV/T System Employed in the Tests

The experimental setup of the proposed PV/T and PV systems installed horizontally in the Engineering building at Sohar University, Oman, is shown in Figure 2a. The horizontal position (while usually PV modules are installed with a tilt angle of 28° depending on the location) was chosen for two main reasons: First, to neutralize the effect of the tilt angle of the PV panel on the rate of received solar radiation. Second, the location of the tested modules allows them to receive sunlight for the longest period of time without hindrance. This installation method is consistent with the method used in many recent studies, such as [5,6,8,15]. Moreover, Figure 2b,c show the constructions of the PV module and cross-section of the PV/T system. The PV modules for all systems are monocrystalline and identically selected to have 100 W, 22.32 V, and 5.94 A rated power, open-circuit voltage, and short-circuit current, respectively. In the PV/T systems, the thermal collectors used were of spiral flow type to gain the maximum possible cooling depending on the results of [6]. These collectors were welded on the PV panels' back, which was coated by a thin layer of silicon oil to prevent air gaps forming between the collectors' walls and the PV.

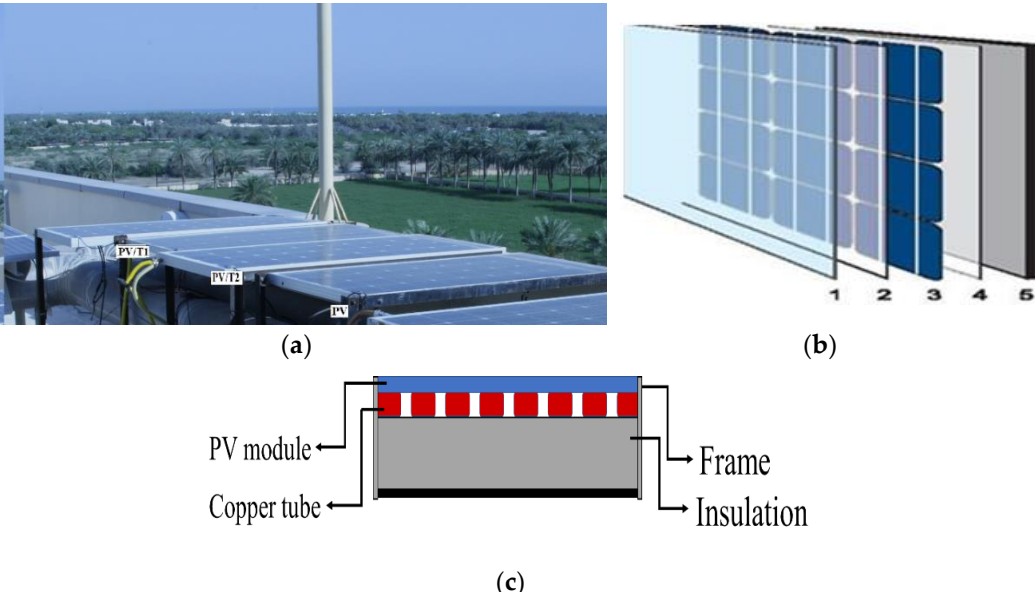

**Figure 2.** (**a**) Experimental setup of PV/T; (**b**) constructions of PV module; (**c**) cross-section of PV/T module.

Two W1209 digital temperature sensors (temperature range: $-50$ to $110\ ^\circ$C, accuracy $\pm0.1\ ^\circ$C) were connected at the PV/T inlet and outlet PV/T collector to measure the temperature via seven segment displays, while two DS18B20 sensors (temperature range: $-55$ to $125\ ^\circ$C, accuracy $\pm0.5\ ^\circ$C) were connected at the PV and PV/T cells to measure the temperatures of the cells. A 1 to 30 liter/minute water flow rate sensor (1/2-inch water flow sensor, model YF-S201, accuracy $\pm10\%$) and a DC water pump (model: FL-2201, 1.6 A, 12 V) were used to circulate the water through the pipe that was used. However, the losses due to the pump consumption have not been considered in this study. Data have been recorded for September and October 2021. The best three days' measurements were recorded and considered from 08:00 AM till 05:00 PM for the period 1st–3rd October 2021. A sensor and data acquisition system were used to measure and record the PV and PV/T current and voltage as shown in Figure 3. Moreover, solar irradiance and ambient temperature were recorded. The recorded measured values are saved in the computer to be analyzed. For data reduction, the data acquisition has been set to take measurements every 30 min.

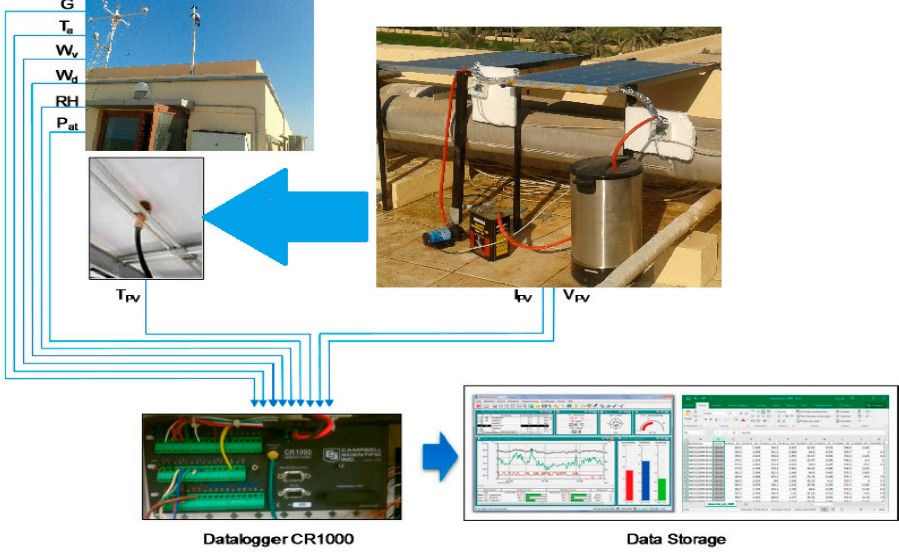

**Figure 3.** Experimental setup and data flow of PV/T and PV systems.

In this study, a comparison between three identical PV modules, two of them with a thermal collector (PV/T) as shown in Figure 2a, is presented. In the first PV/T system, water was used as a cooling fluid. In the second PV/T system, SiC nanofluid (SiC-water) was used as a cooling fluid to transfer the thermal energy (heat) from the solar cells to enhance the PV efficiency and increase the extracted thermal energy.

### 2.4. Performance Evaluation Parameters and Uncertainty

The performance evaluation parameters are illustrated in Table 3. Equations (1)–(5) illustrated the performance parameters used to evaluate the PV/T system. Equations (1) and (2) were used to calculate maximum PV/T electrical and thermal power, respectively. Equations (3)–(5) were used to determine the electrical, thermal, and overall efficiencies, respectively [3].

$$P_{max} = I_{mp} \times V_{mp} \tag{1}$$

$$Q_u = \dot{m}C_p(T_o - T_i) \tag{2}$$

$$\eta_e = \frac{P_{max}}{I_s \times A_{panel}} \tag{3}$$

$$\eta_{th} = \frac{Q_u}{I_s \times A_c} \tag{4}$$

$$(\eta_t) = \eta_t = \eta_{th} + \eta_e = \frac{Q_u + P}{I_s \times A_t} \tag{5}$$

**Table 3.** PV/T performance evaluation parameters formula.

| Parameter | Unit | No. of Equation |
|---|---|---|
| PV/T power | W | Equation (1) |
| Collected heat | W | Equation (2) |
| Electrical efficiency | % | Equation (3) |
| Thermal efficiency | % | Equation (4) |
| PV/T efficiency | % | Equation (5) |

An ultrasonic vibrating mixer (TELSONIC ULTRASONICS CT-I2) was used to achieve an accurate result. The added weights were verified using an accurate digital scale, type METTLER TOLEDO's (US made), that measures up to 0.0001 gram. The thermal conductivity of the prepared nanofluids was measured with a KD2 Pro analyzer scale (ICT International, India). As for the stability of the prepared fluids, it was measured using Nano Zeta-Sizer (ZSN) (GmbH). Each set of experiments and measurements was repeated three times as a way to confirm the repeatability of experiments and reduce measurement uncertainty. Each instrument was calibrated before use, and their accuracy was determined. These values were used to determine the uncertainty, the details of which are listed in Table 3. The following equation [49] shows the total uncertainty of the experiments:

$$e_R = \left[ \left( \frac{\partial R}{\partial V_1} e_1 \right)^2 + \left( \frac{\partial R}{\partial V_2} e_2 \right)^2 + \ldots + \left( \frac{\partial R}{\partial V_n} e_n \right)^2 \right]^{0.5} \tag{6}$$

where $e_R$, $R$, $e_i$, and $\frac{\partial R}{\partial V_1}$ represent the results of uncertainty, independent variable's function, the uncertainty interval in the $n_{th}$ variable, and single variable measured result sensitivity, respectively. Table 4 lists the instruments used and their uncertainties. The total test instrumentations' uncertainty was 1.802, which reveals acceptable accuracy of the measuring devices.

**Table 4.** Instrumentations' uncertainties.

| No. | Measured Variable | Measuring Devise | Uncertainty (%) |
|---|---|---|---|
| 1 | Electrical variables (current and voltage) | Multi-meter | 0.96 |
| 2 | Nanofluid flow rate | Flowmeter | 0.28 |
| 3 | Temperature | Thermocouples | 0.34 |
| 4 | Irradiance | Solar radiation intensity meter | 0.91 |
| 5 | Nanoparticle mass fraction weight | Sensitive weight | 0.001 |
| 6 | Nanofluids density | Density tester | 0.42 |
| 7 | Nanofluids viscosity | Brookfield Programmer Viscometer (Model: LVDV-III Ultra-programmable) | 0.53 |
| 8 | Thermal conductivity and capacity | Hot desk Tps 500 | 0.92 |

## 3. Results and Discussion

### 3.1. Thermophysical Properties

An additive mass fraction of nano-SiC 0.5% was chosen based on the results of [33], as mentioned previously. The effect of temperature change on samples of the prepared suspension was examined. Table 5 shows this effect of the nano-suspension samples at 10 °C phases. The samples were heated from 25 °C to 75 °C, which is the summer operating temperature range of PV/T systems in Sohar.

**Table 5.** The effect of temperature on the samples' thermophysical properties.

| Property / Temperature (°C) | Density (kg/m$^3$) | Viscosity (mPs.s) | Thermal Conductivity Enhancement Rate (%) | Stability (Zeta Potential) |
|---|---|---|---|---|
| 25 | 1.13 | 1.12 | 6.64 | 63 |
| 35 | 1.045 | 1.07 | 6.98 | 61 |
| 45 | 0.98 | 1.01 | 7.67 | 58.7 |
| 55 | 0.94 | 0.985 | 7.98 | 57.3 |
| 65 | 0.89 | 0.955 | 8.16 | 56.5 |
| 75 | 0.85 | 0.923 | 8.72 | 55.8 |

Measurements show that a significant increase occurs when nano-SiC is added to water, reaching 13% in its density. However, the density decreases rapidly with increasing temperatures. Heat causes the fluid to expand and thus to decrease its density, and since the nanofluid has a high conductivity, its expansion speed is higher than that of water, so its density decrease is obvious. The nanofluid density reduced by 12.27% and 21.23% when the fluid operated at 45 °C and 65 °C, respectively. The high density of the nanofluid will cause a high load on the circulating pump, especially at the starting period. However, this load decreases with the increase in the temperature of the circulating fluid. Here, the designer must compromise carefully between the benefits gained from the system's cooling versus the additional electricity consumption during the start of operation.

The viscosity of the nanofluid increases compared to water by about 12% at a temperature of 25 °C, as the measurements show in Table 5. However, this viscosity decreases with increasing temperature, which causes a higher movement of nanoparticles with the expansion of the fluid and the separation of these particles suspended in the solution. The viscosity reduction rates of 4.46%, 12.05, and 17.58% were experienced at 35 °C, 55 °C, and 75 °C operation temperatures, respectively. An increase in viscosity causes a rise in the load applied to the pump, and its reduction reduces the electricity consumption of the latter. Therefore, it is expected that the pump start-up will be difficult, and after the nanofluid acquires heat, the load on it will be significantly reduced.

Increasing the flow rate of the fluid, whether water or nanofluid, causes an increase in the amount of heat absorbed by the PV panel. In fact, this statement is not an absolute but rather limited. After reviewing Equation (2) (Table 3), it is noted that the mass flow rate and the difference in the temperatures entering and leaving the collector are the two factors affecting the amount of heat absorbed, given that the Cp change with the increase in temperature is very limited. So, if the mass flow rate increases dramatically, the temperature difference will decrease. This inverse relationship depends largely on the intensity of the solar radiation, the temperature of the PV panel, the rate of heat transfers between the plate and the coolant, and the collector efficiency.

The increase in the density and viscosity of the base fluid as a result of adding nanoparticles necessitates an increase in the pumping power, which means an increase in electricity consumption. To prevent any confusion and misleading in this matter, since one could understand that the pumping power would be reduced when the flow becomes turbulent, the concept presented by equations in [50] was adopted to estimate the assumed increase in pumping power:

$$Laminar\ flow: \left(\frac{W_{nf}}{W_{bf}}\right) = \left(\frac{\mu_{nf}}{\mu_{bf}}\right) \cdot \left(\frac{\rho_{bf}}{\rho_{nf}}\right)^2 \tag{7}$$

$$Turbulent\ flow: \left(\frac{W_{nf}}{W_{bf}}\right) = \left(\frac{\mu_{nf}}{\mu_{bf}}\right)^{0.25} \cdot \left(\frac{\rho_{bf}}{\rho_{nf}}\right)^2 \tag{8}$$

where $W$ is pumping power, and *nf* means nanofluid while *bf* means base fluid. Table 6 lists the results of Mansour Equations (7) and (8), which mathematically relates the pumping power to the density and viscosity of the used fluid and temperature variations for the studied case. In Table 6, to prevent any misleading in the results, the values listed in the table are not absolute values of pumping power, which change every second with the change in the temperature of the working fluid.

**Table 6.** Laminar and turbulent flow consumption results.

| Flow Type | 25 °C | 35 °C | 45 °C | 55 °C | 65 °C | 75 °C |
|---|---|---|---|---|---|---|
| Laminar flow (Equation (7)) | 0.877 | 0.98 | 1.012 | 1.046 | 1.051 | 1.148 |
| Turbulent flow (Equation (8)) | 0.805 | 0.813 | 0.886 | 0.918 | 0.943 | 0.966 |

Ben et al. (2007) [50] suggested that ($W_{\text{Nanofluid}}/W_{\text{bf}} < 1$) for both types of flow is very preferable, since in this case the electricity consumption will not increase due to the base fluid exchange to nanofluid while maintaining a higher heat transfer rate. Table 6 shows the effect of adding Nano-SiC to water on the pumping power of the system. The results included show that the use of laminar flow causes an increase in the $W_{\text{Nanofluid}}/W_{\text{bf}}$ ratio to above one for operating at a temperature of more than 45 °C. Therefore, it is not recommended to use laminar flow with a nanofluid-cooling PV/T system. In the case of turbulent flow, the $W_{\text{Nanofluid}}/W_{\text{bf}}$ values were all below one. Therefore, it must be emphasized here that this option (the use of turbulent operation) is preferred to be adopted when working with the studied nanofluid for all operating temperatures. The results of this study are fully consistent with what Asadi (2018) [51] has reported.

The thermal conductivity of the nanofluid is greatly increased compared to water due to the high conductivity of the nano-SiC. The conductivity also increases with the increase in the temperature of the nanofluid, as the heat energy gained causes a rise in the movement of nanoparticles and in heat transfer. The nanofluid's thermal conductivity is enhanced by 5.12%, 15.51%, 20.18%, 22.89%, and 31.32% when its temperature is raised from 35 °C, 34 °C, 55 °C, 65 °C, and 75 °C, respectively. The main task of the nanofluid is heat transfer, and the higher the temperature difference between the fluid and the surface

of the PV panel, the greater the amount of heat transferred and the better the thermal performance of the system.

The zeta potentiometer is a measure of fluid stability and the survival of nanoparticles suspended through it. Whenever this scale is higher than 40, the stability is high and acceptable. The measurements in Table 5 show acceptable stability at all studied temperatures, even if the stability decreased by increasing the temperature from 25 °C to 75 °C by 11.42%. The accepted stability comes from two factors considered when nano-SiC is selected. First, the size of particles 25–35 nm (Table 5) is small, which encourage suspension after good mixing. Secondly, the added mass fraction was limited to ensure good suspension. As the temperature of the fluid increases, the movement of nanoparticles increases, their collision with each other increases, and the possibility of their agglomeration becomes greater.

### 3.2. Experimental Results

In this paper, two photovoltaic modules with a thermal collector (PV/T) and a without thermal collector (PV) were tested in term temperature and solar irradiance. Electrical quantities such as voltage, current, and power were measured and recorded. Moreover, PV and PV/T temperature were measured. Furthermore, the water inlet and outlet temperature were measured and recorded as shown in Figure 4a–c. The measurement was conducted in September and October 2020. However, the best three days, 1–3 October, were selected for analysis, discussion, and comparison.

The PV/T temperature is lower than PV temperature due to the cooling, which is reflected on voltage and power significantly compared with current. The increase of temperature has an insignificant increase of current and a significant increase of voltage [1,45,52–58]. Figure 4a shows the improvement of PV/T voltage compared with conventional PV due to the cooling. The PV/T voltage curve shows strange spikes caused by the instantaneous increase in the intensity of solar radiation. Since the measurement here was in min, these spikes are visible, while they are not visible in the other figures where the average readings are plotted. The PV/T voltage, power, and efficiency were increased as shown in Figure 4b compared with the PV module. It is clear that the PV/T voltage improvement reflected positively on power and efficiency. Figure 4c shows inlet and outlet temperature increment. It was found that, after 09:20 AM, the inlet and outlet temperatures increased more than the ambient temperature, where the outlet temperature was higher compared with the inlet temperature. At 12:10 PM, the difference of the two previews' temperatures was increased, the cooling became more effective, and more heat was extracted from the module. To make sure the cooling is effective through the whole day from sunrise to sunset, it is advisable to have a large tank size. Moreover, it was found that from 10:00 AM to 05:00 PM, the PV/T temperature reduced significantly compared with conventional PV. The PV/T curve is very much fluctuating in the figure due to the difference between the coolant inlet and exit temperatures and the instantaneous conditions of the solar radiation intensity received by the PV module. This oscillation can be eliminated by adding a phase-change material (PCM) attached to the collector, as declared by Al-Waeli et al. [44], Qui et al. [59], and Fiorentini et al. [60].

Figure 5 shows the change in the electrical and thermal efficiencies of the studied systems over time. It was noted that the electrical efficiency of the nanofluid-cooled PV/T system is higher than the other two cases. The increase in the average electrical efficiency for both water-cooled and nanofluid-cooled PV/T systems was 12.6% and 25.3%, respectively, compared to the electrical efficiency of standalone PV. The recycling of nano-SiC in the PV/T system caused a superior cooling effect compared to water cooling. The average thermal efficiency of the two water-cooled and nano-SiC systems reached 17.83% and 35.02%, respectively, with the latter being superior by 98.6%. The high thermal conductivity of SiC nanofluid absorbs more heat than water, causing the cooling effect to be approximately doubled.

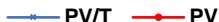

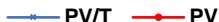

**Figure 4.** *Cont.*

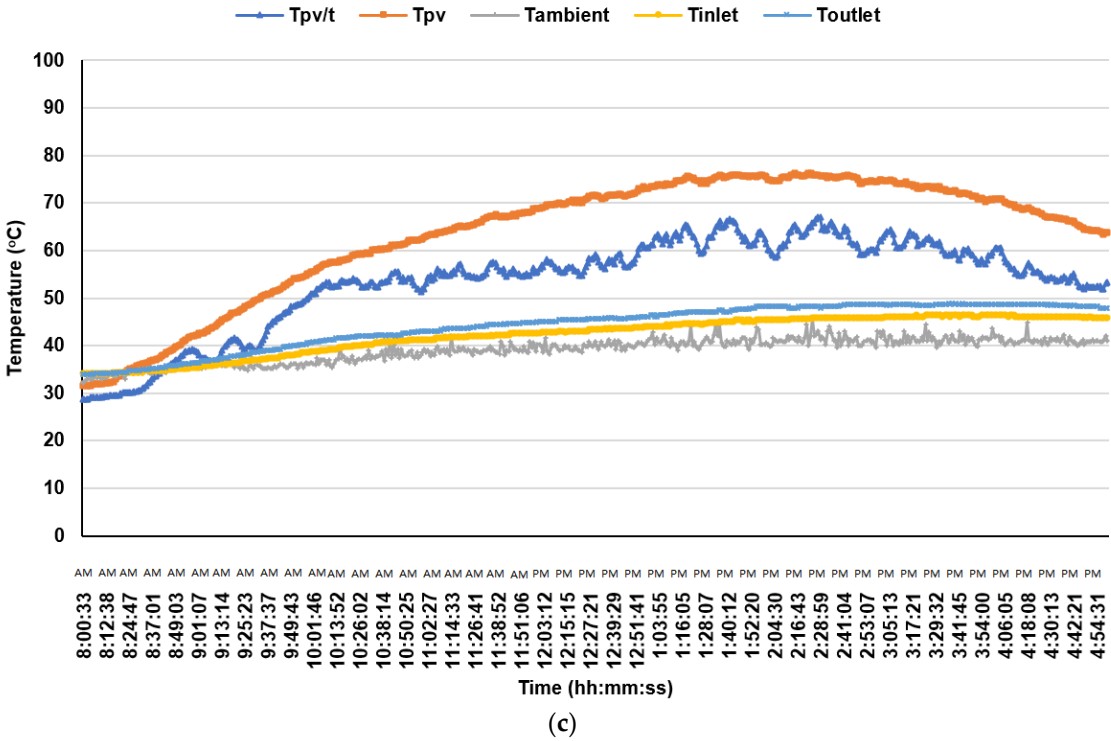

**Figure 4.** (**a**) PV and PV/T voltages (minutes), (**b**) PV and PV/T current, voltage, and power measurements (every 30 min), and (**c**) ambient, inlet, outlet, PV/T, and PV temperatures.

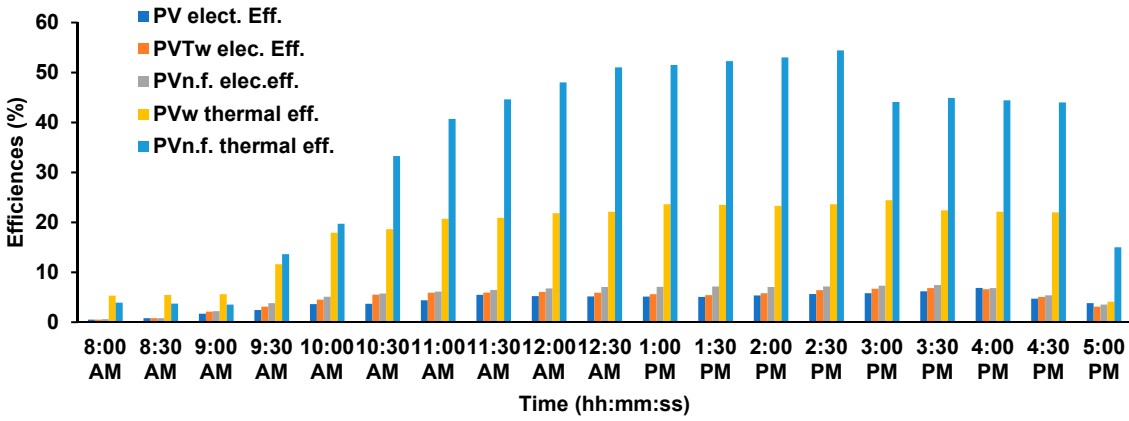

**Figure 5.** PV and PV/T efficiencies variations with time.

### 3.3. Comparison with Literature

In this section, the study results were compared with literature results. Three comparisons were conducted with respect to different cooling methods (Table 7), with respect to different nanofluids (Figure 6) and with respect to similar studies that used SiC nanoparticles (Figure 7), respectively. It is worth mentioning that, despite the differences of the investigated systems and locations, the efficiencies gave an indication about the cooling level. Table 7 illustrates the comparison of PV/T electrical and thermal efficiency in terms of cooling methods. It was found that the thermal efficiency is relatively high and is consistent with literature results. Moreover, air cooling has the lowest efficiencies ($\eta_e = 7.7\%$, $\eta_{th} = 28\%$), but nanofluid cooling has the highest efficiencies ($\eta_e = 13.14\%$, $\eta_{th} = 68.22\%$).

**Table 7.** Comparison of PV/T based on cooling method.

| Ref. | Country | Thermal Efficiency (%) | Electrical Efficiency (%) | Cooling Method |
|------|---------|------------------------|---------------------------|----------------|
| [53] | China | 28 | 7.7 | Air |
| [54] | Canada | 48 | 16.5 | Air |
| [55] | Bangladesh | 30 | 9.25 | Water |
| [56] | Italy | 62 | 13.19 | Water |
| [57] | China | 64.4 | 11.8 | Water-air |
| [58] | China | 76 | 17 | Water-air |
| [59] | UK | 59 | 8.7 | PCM |
| [60] | Australia | 45 | 9 | PCM |
| [33] | Malaysia | 67 | 13.5 | Nanofluid |
| [13] | Iran | 33 | 17 | Nanofluid |
| [44] | Malaysia | 72 | 13.7 | Nanofluid-Nano/PCM |
| [61] | Iran | 47 | - | Nanofluid-Nano/PCM |
| Current study | Oman | 68.22 | 13.14 | SiC Nanofluid |

Figure 5 shows the efficiencies as a compound bar where both electrical and thermal efficiencies are compared considering different nanoparticles [13,30,33,36,62–68]. Despite the differences of the investigated panel, technology, systems, rating, and locations, the efficiencies gave an indication about the efficiencies related to the cooling nanofluid. The CuO, Al2O4, and SiC show the highest efficiencies of 88.78%, 81.00, and 80.50%, respectively. Moreover, the current study results are consistent with the literature results. The thermal efficiencies are more effected by nanofluid cooling compared with electrical efficiencies.

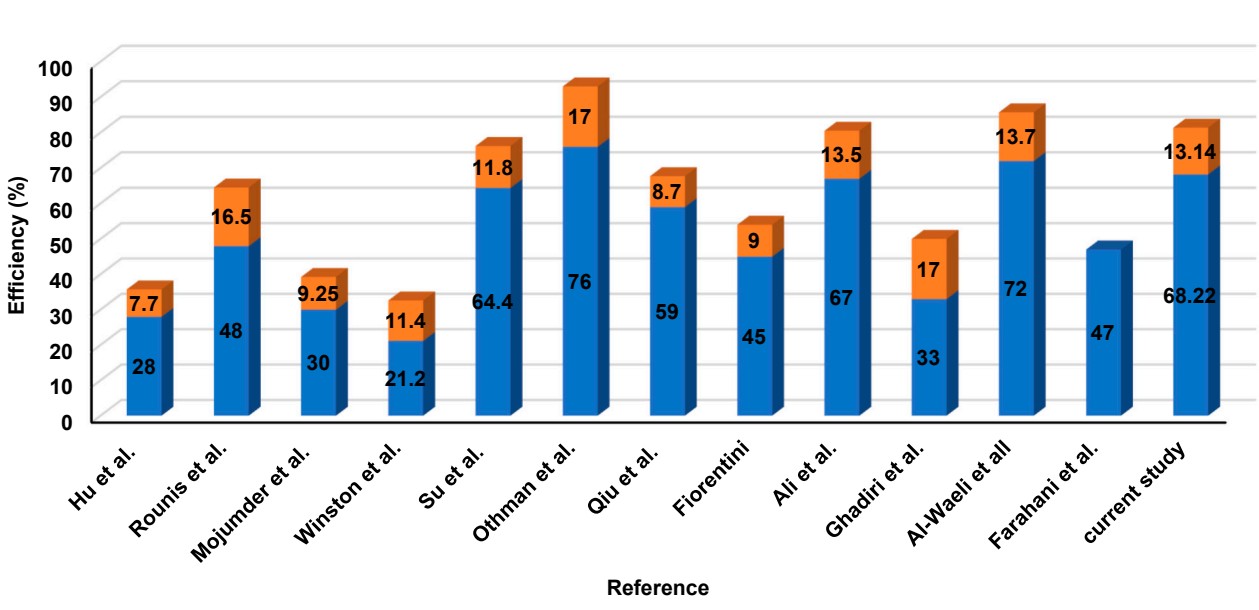

**Figure 6.** Comparison of PV/T based on used nanofluid.

Figure 6 shows a spiral comparison of thermal and electrical efficiencies of PV/T systems using SiC nanofluid cooling [33,44,63,69–72]. Figure 6a,b illustrate the consistency

of the tested system efficiencies with other systems in the literature, which verified the SiC nanofluid's cooling superiority.

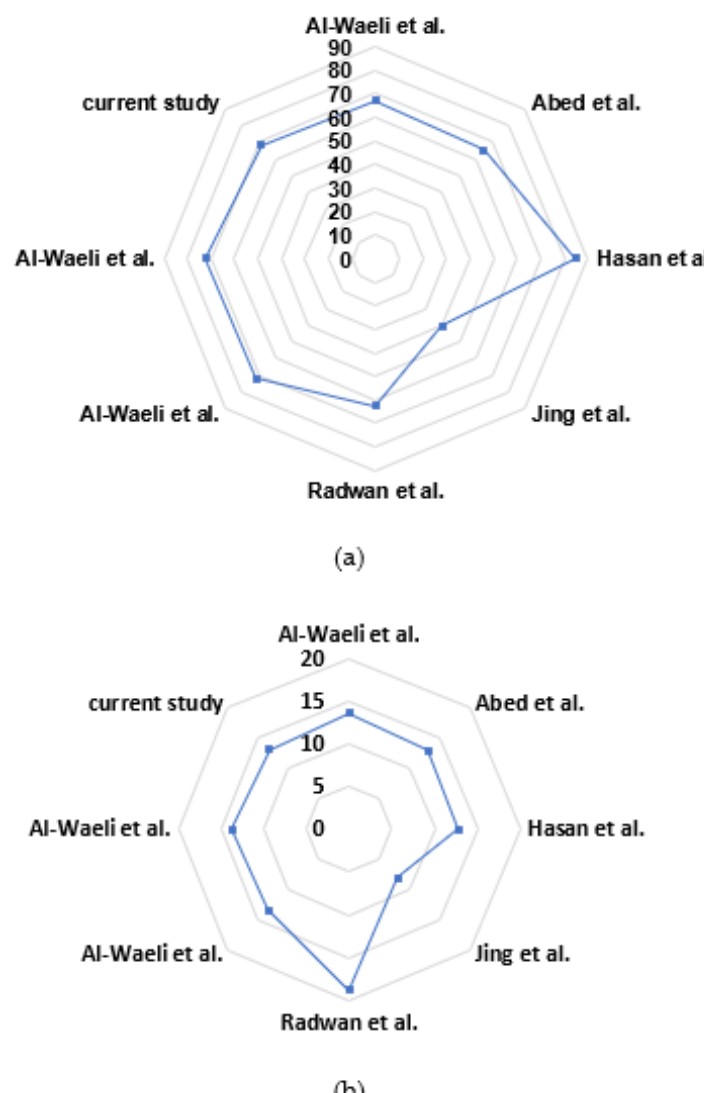

**Figure 7.** Comparison of PV/T with similar nanofluid (SiC): (**a**) thermal efficiency; (**b**) electrical efficiency.

## 4. Conclusions

For the purpose of preparing ready-made specifications for designers and decision makers regarding the quality of the nanofluid to be used in PV/T systems in the Omani city of Sohar, the PV/T system was tested experimentally using silicon carbide and water. Nano-SiC with a mass fraction of 0.5% was added to water with a surfactant. The thermophysical properties of the prepared nanofluid at different operating temperatures were investigated. At 25 °C, the thermal conductivity, density, and viscosity of the nanofluid increased by 6.64%, 13%, and 12%, respectively. The effect of the apparent increase in the density and viscosity of the nanofluid compared to the base fluid on the pumping power was investigated. The study showed that the work of the system with the laminar flow will cause losses in pumping power, while the use of turbulent flow reduces the required power. The electrical efficiency has increased up to 25.3% compared to a conventional PV module as a result of proper cooling. Thermal efficiency has also increased by 98.6% compared to the water-cooled PV/T system. The results of the current study with their counterparts in the literature, in terms of cooling methods and nanoparticles and similar studies that used SiC nanofluid, were compared. The results of the comparison show high potential for a nanofluid prepared by the method used in this study.

**Author Contributions:** Conceptualization, H.A.K. and M.T.C.; methodology, H.A.K.; validation, A.H.A.A.-W.; H.J.; A.I.; formal analysis, A.H.A.A.-W. and H.J.; investigation, H.A.K.; resources, H.A.K., M.T.C. and A.H.A.A.-W.; data curation, H.A.K. and A.H.A.A.-W.; writing—original draft preparation, H.A.K., M.T.C. and A.I.; writing—review and editing, H.A.K. and M.T.C.; supervision, K.S. All authors have read and agreed to the published version of the manuscript.

**Funding:** The research leading to these results has received Research Project Grants Funding from the Research Council of the Sultanate of Oman, Research Grant Agreement No. ORG SU EI 11 010.

**Institutional Review Board Statement:** Not applicable.

**Informed Consent Statement:** Not applicable.

**Data Availability Statement:** Not applicable.

**Acknowledgments:** The authors acknowledge Research Project Grants Funding from the Research Council of the Sultanate of Oman, Research Grant Agreement No. ORG SU EI 11 010.

**Conflicts of Interest:** The authors declare that there is no conflict of interests regarding the publication of this paper. We the authors declare no affiliations with or involvement in any organization or entity with any financial interest (such as honoraria; educational grants; participation in speakers' bureaus; membership, employment, consultancies, stock ownership, or other equity interest; and expert testimony or patent licensing arrangements), or non-financial interest (such as personal or professional relationships, affiliations, knowledge, or beliefs) in the subject matter or materials discussed in this manuscript.

## Nomenclature

| | |
|---|---|
| $A_C$ and $A_{module}$ | Collector and PV areas ($m^2$) |
| $Cp$ | Water heat capacity (J/(K kg)) |
| $G$ | Solar irradiance ($W/m^2$) |
| $GS$ | Global solar radiation ($W/m^2$) |
| $I_{SC}$ and $I_{mp}$ | Short circuit and maximum point currents (A) |
| $M_F$ | Mass flow (kg/h) |
| PV | Photovoltaic |
| PVT | Photovoltaic/Thermal |
| $P_{rated}$ and $P_{mp}$ | Rated and maximum point powers (W) |
| $T_{ambient}$ | Ambient temperature (°C) |
| $T_C$ | Cell temperature (°C) |
| $T_{in}$ and $T_{out}$ | Inlet and outlet temperature (°C) |
| $V_{OC}$ and $V_{mp}$ | Open circuit and maximum point voltages (V) |
| $W_R$ | Uncertainty |
| $\eta_{electrical}$ and $\eta_{thermal}$ | Electrical and thermal efficiencies (%) |

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
