# Peer review of "Effect of Temperature on the Electrical and Thermal Behaviour of a Photovoltaic/Thermal System Cooled Using SiC Nanofluid: An Experimental and Comparison Study"

_sustainability, doi:10.3390/su141911897_

Round 1
Reviewer 1 Report
Please see attachment
Paper is good and can subject to modification as per comments.

Reviewer 2 Report
The review comments for the manuscript, 'Effect of Temperature on the Electrical and Thermal Behaviour of Photovoltaic/Thermal System Cooled Using SiC Nanofluid: An Experimental and Comparison Study', are given below,
1. The authors presented a PV/T systemwith SiC cooler. The work is really good and presented well, still it needs few changes for betterment.
2. Add nomenclature for the symbols used and abbreviations used.
3. Also add the organization of the manuscript at the end of Introduction section.
4. Why the panels are kept flat/horizontal in experimental setup? What is the impact of it and why the research latitude is mentioned?
5. What about the error/uncertainity analysis? 10.1016/j.desal.2017.11.007, 10.1016/j.enconman.2018.05.011 for the economic analysis and payback period.
6. References need to be updated with recent ones.
7. Authors need to check the all the figures and numbers. Few figures were not cited in the text.
8. So many typos are there throughout the manuscript and it must be checked.
9. Abstract and conclusion must be brief and precise.
Reviewer 3 Report
The authors tested a PV/T system Cooled Using SiC Nanofluid in Oman and compared the results with literature results. The cooling medium contained 0.5 wt.% of nanoparticles and tested in terms of thermophysical properties, performance parameters, and efficiency.
The authors must address the following issues in a revised version of their manuscript:
“At 25°C, the authors found an increase in the nanofluid’s thermal conductivity, density, and viscosity up to 6.64%, 13%, and 12%, respectively”. These are reasonable findings.
“However, when examining the effect of increasing density and viscosity to the pumping power, it was found that using turbulent flow reduces the required pumping force and vice versa for the case of laminar flow”.
This sentence is not accurate. Moreover, it is misleading, since one could understand that the pumping power would be reduced when the flow becomes turbulent, which is not correct.
Table 6 is misleading: These are not absolute values of pumping power to be compared between laminar and turbulent flow! Which is the parameter compared here???
English need significant improvement, preferably by a native English speaker.
For example:
“2.3 The used PV/T system” should be: “2.3 The PV/T system employed in the tests”
Line 218: 4565 should be 45
Line 236: Laminar
Lines 236-237 Please number these equations.
Also, these are just approximations. They should not be used here to draw conclusions on the pumping power. Instead, the pumping power should be measured and this would be a valid contribution of your paper.
Figure 5a: the voltage curve for the PV/T presents peculiar spikes that must be explained.
Figure 5c: the PV/T temperature curve is very much fluctuating: you must explain this fact.
Figure 6 is not good to convey the correct meaning: For example Khanjari et al. present a very low electrical efficiency. What type of panel is it? The same with Sardar.
Round 2
Reviewer 3 Report
Can be published in the revised version
Author Response
Done. Thank you for the valuable comments which enrich the manuscript.
